# Rosenblatt’s First Theorem and Frugality of Deep Learning

**DOI:** 10.3390/e24111635

**Published:** 2022-11-10

**Authors:** Alexander Kirdin, Sergey Sidorov, Nikolai Zolotykh

**Affiliations:** 1Institute of Information Technologies, Mathematics and Mechanics, Lobachevsky State University, 603022 Nizhni Novgorod, Russia; 2Institute for Computational Modelling, Russian Academy of Sciences, Siberian Branch, 660036 Krasnoyarsk, Russia

**Keywords:** complexity, classification, shallow network, elementary perceptron, deep network, travel maze problem

## Abstract

The Rosenblatt’s first theorem about the omnipotence of shallow networks states that elementary perceptrons can solve any classification problem if there are no discrepancies in the training set. Minsky and Papert considered elementary perceptrons with restrictions on the neural inputs: a bounded number of connections or a relatively small diameter of the receptive field for each neuron at the hidden layer. They proved that under these constraints, an elementary perceptron cannot solve some problems, such as the connectivity of input images or the parity of pixels in them. In this note, we demonstrated Rosenblatt’s first theorem at work, showed how an elementary perceptron can solve a version of the travel maze problem, and analysed the complexity of that solution. We also constructed a deep network algorithm for the same problem. It is much more efficient. The shallow network uses an exponentially large number of neurons on the hidden layer (Rosenblatt’s *A*-elements), whereas for the deep network, the second-order polynomial complexity is sufficient. We demonstrated that for the same complex problem, the deep network can be much smaller and reveal a heuristic behind this effect.

## 1. Introduction

Rosenblatt [1] studied elementary perceptrons (Figure 1). *A*- and *R*-elements are the classical linear threshold neurons. The *R*-element is trainable by the Rosenblatt algorithm, while the *A*-elements should represent a sufficient collection of features.

Rosenblatt assumed no restrictions on the choice of the *A*-elements. He proved that the elementary perceptrons can separate any two non-intersecting sets of binary images (Rosenblatt’s first theorem in [1]). The proof was very simple. For each binary image *x*, we can create an *A*-element $A_{x}$ that produces output 1 for this image and 0 for all others. Indeed, let the input retina have *n* elements and $x=(x_{1},\ldots ,x_{n})$ be a binary vector ($x_{i}=0$ or 1) with *k* non-zero elements. The corresponding *A*-element $A_{x}$ has input synapses with weights $w_{i}=1/k$ if $x_{i}=1$ and $w_{i}=-1/k$ if $x_{i}=0$. For an arbitrary binary image *y*,
$$\sum w_{i}y_{i}\leq 1,$$
and this sum is equal to 1 if and only if y=x. The threshold for the $A_{x}$ output can be selected as $1-\frac{1}{2k}$. Thus,
(1)$$OutA_{x}\left(y\right)=\left\{\begin{array}{cc} 0, & \mathrm{if}\;\sum w_{i}y_{i}<1-\frac{1}{2k};\\ 1, & \mathrm{if}\;\sum w_{i}y_{i}\geq 1-\frac{1}{2k}. \end{array}\right.$$

The set of neurons $A_{x}$ created for all binary vectors *x* transforms binary images into the vertexes of the standard simplex in $R^{2^{n}}$ with coordinates $OutA_{x}\left(y\right)$ (Equation 1). Any two non-intersecting subsets of the standard simplex can be separated by a hyperplane. Therefore, there exists an *R*-element that separates them. According to the convergence theorem (Rosenblatt’s fourth theorem in [1]), this *R*-element can be found by the perceptron learning algorithm (a simple relaxation method for solving of systems of linear inequalities).

Thus, Rosenblatt’s first theorem is proven:

**Theorem** **1.**
*An elementary perceptron can separate any two non-intersecting sets of binary images.*


Of course, selection of the *A*-elements in the proposed form *for all* $2^{n}$ binary images is not necessary in the realistic applied classification problems. Sometimes, even an empty hidden layer can be used (the so-called linearly separable problems). Therefore, together with Rosenblatt’s first theorem, we obtain a problem about the reasonable (if not optimal) selection of *A*-element. There are many frameworks for approaching this question, for example, the general feature selection algorithms: we generate (for example, randomly, or with some additional heuristics) a large set of *A*-elements that is sufficient for solving the classification problem, and then select the appropriate set of features using different methods. For the bibliography about feature selection, we refer to recent reviews [2,3,4].

The Minsky and Papert book *Perceptron* [5] was published seven years later than Rosenblatt’s book. They started from the restricted percetrons and assumed that each *A*-element has a bounded receptive field (either by a pre-selected diameter or by the bounded number of inputs). Immediately, instead of Rosenblatt’s omnipotence of unrestricted perceptrons, they found that the elementary perceptron with such restrictions cannot solve some problems, such as the connectivity of the image or the parity of the number of pixels in it.

Minsky and Papert proposed various definitions of restrictions. The first was the definition of the local order of a predicate. A predicate Ψ computed by an elementary perceptron is *conjunctively local of order k* if each *A*-element has only *k* inputs (depends only on *k* points) and
$$\Psi =\left\{\begin{array}{cc} 1 & \mathrm{if}\;\mathrm{all}\;\mathrm{outputs}\;\mathrm{of}\;A-\mathrm{elements}\;y_{i}=1,\\ 0 & \mathrm{otherwise}. \end{array}\right.$$

Minsky and Papert studied flat binary image predicates. The first theorem about a limited perceptron stated that if *S* is a square lattice, then the *connectivity* of an image is not a conjuctively local predicate of any order. Then they published a much deeper theorem without the condition of conjunctive form but for the bounded number of inputs of *A*-elements: the only topologically invariant predicates of finite order are functions of the Euler characteristic of the image (Michael Paterson Theorem). Then they mentioned: “It seems intuitively clear that the abstract quality of connectedness cannot be captured by a perceptron of finite order because of its inherently serial character: one cannot conclude that a figure is connected by any simple order-independent combination of simple tests.” The serial Turing machine serial algorithm for the connectivity test was presented with the evaluation of the necessary memory (the number of squares of the Turing tape): “For any ε there is a 2-symbol Turing machine that can verify the connectedness of a figure *X* on any rectangular array *S*, using less than $\left(1+\varepsilon \right)\log_{2}\left|S\right|$ squares of tape”. That is not very far from the deep multilayer perceptons with depth $∼\log_{2}\left|S\right|$, but this step was not executed.

Minsky and Papert’s results were generalized to more general metric spaces and graphs [6]. The heuristic behind these results is quite simple: if a human cannot solve the problem immediately at a glance and needs to apply some sequential operations such as counting pixels or following a tangled path, then this problem is not solvable by a restricted elementary perceptron.

At the same time, we can expect that the unrestricted elementary perceptron can solve this problem but at the cost of great (exponential?) complexity. Multilayer (“deep”) networks are expected to solve these problems without an explosion of complexity. In that sense, deep networks should be simpler than shallow networks for the problems that cannot be solved by restricted elementary perceptrons and require (from humans) a combination of parallel and sequential actions.

As we can see, there is no contradiction between the omnipotence of unrestricted perceptrons and the limited abilities of perceptrons with limitation. Moreover, there is a clear heuristic that allows us to find the problems of “inherently serial character” (following the Minsky and Papert comment). A human cannot solve such problems with one glance and needs some serial actions, such as following tangled paths.

In this note, we demonstrate the relative simplicity of deep solvers on a version of the well-known travel maze problem (Figure 2). This geometric problem is closely related to the connectivity problem and has been used for benchmarking in various areas of machine learning (see, for example, [7]). The simplicity of the deep solution for this problem is not a miracle “because of its inherently serial character”. A human solves it following the paths along their length *L*.

For formal analysis of the travel maze problem, we need to represent the paths on a discrete retina of *S*-elements (Figure 1). Then, to implement the logic of the proof of Rosenblatt’s first theorem, each *A*-element should be an indicator element for a possible path. For each guest-delicacy pair in Figure 2a or Figure 2b, an elementary perceptron must be created that returns 1 if there is a path from this guest to this delicacy, and 0 if there are no such paths. Thus, $n^{2}$ elementary perceptrons should be created. We can easily combine them in a shallow network with $n^{2}$ outputs. To finalize the formal statement, we should specify the set of the possible paths. In our work, we select a very simple specification without loops, steps back or non-transversal intersections of paths (Figure 2b).

## 2. Formal Problem Statement

Consider the following problem. There are *n* people, each of whom owns a single object from the set {1,2,…,n}, and different people that own different objects. This correspondence between people and objects can be drawn as a diagram consisting of *n* broken lines, each of which contains *L* links (Figure 3). Each stage of the diagram consists of *n* links and can be encoded by a permutation or, equivalently, by a permutation matrix in a natural way. Namely, if an edge is drawn from node *i* to node π(i)(i=1,2,…,n), then the permutation can be represented in the form
$$\pi =\left(\begin{array}{cccc} 1 & 2 & \cdots & n\\ \pi (1) & \pi (2) & \cdots & \pi (n) \end{array}\right)$$
or by the permutation matrix $P=(p_{i,j}),$ where
$$\begin{array}{c} p_{i,j}=\left\{\begin{array}{cc} 1, & \mathrm{if}\;j=\pi (i)\\ 0, & \mathrm{if}\;j\neq \pi (i). \end{array}\right. \end{array}$$
If the permutation matrix $X_{i}$(i=1,2,…,L) corresponds to the *i*-th stage, then the product $X_{1}\cdot X_{2}\cdot \ldots \cdot X_{L}$ is the permutation matrix again, and it defines the correspondence “person–object”. It is required to construct a shallow (fully connected) neural network that determines the correspondence “person–object” from the diagram.

We formally model a simplified version of the travel maze problem as follows. Each travel maze problem is an *L*-tuple $\left(f_{1},\ldots ,f_{L}\right)$ of bijective functions (permutations) from {1,…,n} to {1,…,n}. Denote by F(n,L) the set of such *L*-tuples. The composition $f_{1}\cdot \ldots \cdot f_{L}$ of these functions from an *L*-tuple is also a bijective function {1,…,n} to {1,…,n} that one-to-one associates objects with persons. Thus, the set of all simplified versions of the travel maze problem is the set F(n,L). In addition, the set F(n,L) is a union of classes of *L*-tuples, each of which corresponds to the same product $f_{1}\cdot \ldots \cdot f_{L}.$

When building the neural network for a travel maze problem, we want each neuron of the neural network to have Boolean values. In particular, each neuron of the next layer is a Boolean function of the neurons of the previous layer.

## 3. Shallow Neural Network Solution

Arrange all the n! permutations $S_{n}=\left\{\pi_{0},\pi_{1},\ldots ,\pi_{n!-1}\right\}.$ Then, $M=\{P_{0},P_{1},\ldots ,P_{n!-1}\}$ is the set of the corresponding n×n permutation matrices.

We denote the entries of the matrix $P_{k}$ by $p_{i,j}^{\left(k\right)},$ where
$$p_{1,\pi_{k}\left(1\right)}^{\left(k\right)}=p_{2,\pi_{k}\left(2\right)}^{\left(k\right)}=\ldots =p_{n,\pi_{k}\left(n\right)}^{\left(k\right)}=1,$$
and the other entries are equal to 0.

Let an *L*-tuple $\left(X_{1},X_{2},\ldots ,X_{L}\right)$ ($X_{i}\in M$ for all i=1,…,L) be an *L*-tuple (a word of length *L*) over the set of permutation matrices *M*. The number of such permutation matrices is n!, and the number of *L*-tuples with elements from *M* is ${\left(n!\right)}^{L}$. Consider all such words arranged by the lexicographical order. For each word $W_{j}=\left(X_{1},X_{2},\ldots ,X_{L}\right)$ with the number *j*$\left(j=0,1,\ldots ,{\left(n!\right)}^{L}-1\right)$, we assign the same number to the product $X_{1}\cdot X_{2}\cdot \ldots \cdot X_{L}=P_{t_{j}}$. The matrix $P_{t_{j}}$ is also an n×n permutation matrix.

Entries of matrices $X_{1},X_{2},\ldots ,X_{L}$ are inputs of the neural network (see Figure 4). Each input corresponds to an input neuron (*S*-element, Figure 1). An inner layer *A*-neuron $y_{j}$ corresponds to the *L*-tuple $W_{j}=\left(X_{1},X_{2},\ldots ,X_{L}\right)$ having the same number *j*. The neuron $y_{j}$ should give the output signal 1 if the input vector is $W_{j}$ and output 0 for all other ${\left(n!\right)}^{L}-1$ possible input vectors. Other input vectors are impossible in our settings (Figure 4). Each matrix element of every permutation matrix $X_{i}$ is either 0 or 1; therefore, the *L*-tuples star of output connections of the inner neuron can be coded as a 0–1 sequence, that is a vertex of the $Ln^{2}$-dimensional unit cube. (Apparently, there are more vertices than *L*-tuples of permutation matrices). This cube is a convex body, and each vertex can be separated from all other vertices by a linear functional. In particular, for each *j*, we can find a linear functional $l_{j}$ such that $l_{j}\left(W_{j}\right)>1/2$ and $l_{j}\left(W_{k}\right)<1/2$ (k≠j, $k,j=1,\ldots ,{\left(n!\right)}^{L}$). Here we, with some abuse of language, use the same notation for the tuple $W_{j}$ and the correspondent vertex of the cube (a 0–1 sequence of the length $Ln^{2}$). Thus, each inner neuron $y_{j}$ can be chosen in the form of the linear threshold element with the output signal (compare to (Equation 1) and Theorem 1):$$y_{j}\left(W\right)=h\left(l_{j}\left(W\right)-1/2\right),$$
where *h* is the Heaviside step function. We use for the output of the neuron $y_{j}$ the same notation $y_{j}$.

The structural difference of the shallow network (Figure 4) for the travel maze problem from the elementary perceptron (Figure 1) is the number of neurons in the output layer. For the travel maze problem, the answer is the permutation matrix with $n^{2}$ 0–1 elements. The inner layer neuron $y_{j}$ detects the *L*-tuple of one-step permutation matrices $W_{j}=\left(X_{1},X_{2},\ldots ,X_{L}\right)$. When this input vector is detected, $y_{i}$ sends the output signal 1 to the output neurons connected with it. For all other input vectors, it keeps silent. The output neurons are simple linear adders. The output neurons $z_{qr}$ are labelled by pairs of indexes, q,r=1,…,n. The matrix of outputs is the permutation matrix from the start to the end of the travel. The structure of the output connections of $y_{i}$ is determined by the input *L*-tuple $W_{j}=\left(X_{1},X_{2},\ldots ,X_{L}\right)$: the connection from $y_{j}$ to $z_{qr}$ has weight 1 if the corresponding entry ${\left(P_{t_{j}}\right)}_{qr}=1$ and is 0 if ${\left(P_{t_{j}}\right)}_{qr}=0$. (Recall that $P_{t_{j}}=X_{1}\cdot X_{2}\cdot \ldots \cdot X_{L}$).

Thus, the neuron $y_{j}$ corresponds to our problem answer. Let us represent the network functioning in more detail with explicit algebraic presentations. All the inputs and outputs are Boolean (0–1) variables. We use the standard Boolean algebra notations. In particular, $\bar{x}=1-x.$
$$\begin{array}{ccc} y_{0} & ⟶ & \overset{L}{\overbrace{P_{0}\cdot \ldots \cdot P_{0}\cdot P_{0}}}=P_{t_{0}},\\ y_{1} & ⟶ & P_{0}\cdot \ldots \cdot P_{0}\cdot P_{1}=P_{t_{1}},\\ & \ldots & \\ y_{n!-1} & ⟶ & P_{0}\cdot \ldots \cdot P_{0}\cdot P_{n!-1}=P_{t_{n!-1}},\\ y_{n!} & ⟶ & P_{0}\cdot \ldots \cdot P_{1}\cdot P_{0}=P_{t_{n!}},\\ & \ldots & \\ y_{2n!-1} & ⟶ & P_{0}\cdot \ldots \cdot P_{1}\cdot P_{n!-1}=P_{t_{2n!-1}},\\ & \ldots & \\ y_{{\left(n!\right)}^{L}-1} & ⟶ & P_{n!-1}\cdot \ldots \cdot P_{n!-1}\cdot P_{n!-1}=P_{t_{{\left(n!\right)}^{L}-1}} \end{array}$$

Thus, if $y_{j}$ is a neuron of the inner layer, then it corresponds to the product
$$P_{t_{j}}=P_{a_{j,L-1}}\cdot P_{a_{j,L-2}}\cdot \ldots \cdot P_{a_{j,1}}\cdot P_{a_{j,0}},$$
where
$$j=a_{j,L-1}{\left(n!\right)}^{L-1}+a_{j,L-2}{\left(n!\right)}^{L-2}+\ldots +a_{j,1}\left(n!\right)+a_{j,0}$$
is the expansion of *j* in the base n!.

We need
$$y_{j}\left(X_{1},X_{2},\ldots ,X_{L}\right)=1\;\;⟺\;\;\left(X_{1},X_{2},\ldots ,X_{L}\right)=\left(P_{a_{j,L-1}},P_{a_{j,L-2}},\ldots ,P_{a_{j,0}}\right).$$

Denote
$$I_{k}=\left\{j:\;t_{j}=k\right\},\;\left(k=0,\ldots ,n!-1\right),$$
$$M_{ij}=\left\{k:\;\pi_{k}\left(i\right)=j\right\}=\left\{k:\;p_{i,j}^{\left(k\right)}=1\right\}.$$
Note that $|M_{ij}|=\left(n-1\right)!$.

Each neuron $y_{j}$ of the inner layer for $j\in I_{k}$ corresponds to the same product $P_{k}$:$$y_{j}=x_{1,\pi_{a_{j,L-1}}\left(1\right)}^{\left(1\right)}\cdot x_{2,\pi_{a_{j,L-1}}\left(2\right)}^{\left(1\right)}\cdot \ldots \cdot x_{n,\pi_{a_{j,L-1}}\left(n\right)}^{\left(1\right)}\cdot x_{1,\pi_{a_{j,L-2}}\left(1\right)}^{\left(2\right)}\cdot x_{2,\pi_{a_{j,L-2}}\left(2\right)}^{\left(2\right)}\cdot \ldots \cdot x_{n,\pi_{a_{j,L-2}}\left(n\right)}^{\left(2\right)}\cdot$$
$$\ldots \cdot x_{1,\pi_{a_{j,0}}\left(1\right)}^{\left(L\right)}\cdot x_{2,\pi_{a_{j,0}}\left(2\right)}^{\left(L\right)}\ldots x_{n,\pi_{a_{j,0}}\left(n\right)}^{\left(L\right)}\cdot \prod_{\left(\alpha ,\beta \right)\neq (i,\pi_{a_{j,s}}\left(i\right))}\bar{x_{\alpha ,\beta }^{\left(\gamma \right)}}.$$
The third level neurons $z_{ij}$ form the matrix $Z=(z_{ij})$ that is the answer to this problem:(2)$$\begin{array}{c} z_{i,j}=\underset{s\in \underset{k\in M_{ij}}{⋃}I_{k}}{⋁}y_{s},\;\left(i,j=1,\ldots ,n\right). \end{array}$$
Since $|M_{ij}|=\left(n-1\right)!,$$|I_{k}{|=\left(n!\right)}^{L-1},$ then the right-hand side in the equality (Equation 2) contains exactly $\left(n-1\right)!\cdot {\left(n!\right)}^{L-1}$ terms $y_{s}.$

**Theorem** **2.**
*For travel maze problems F(n,L), there is a shallow neural network that has a depth of 3,*

$$\left(L+1\right)n^{2}+{\left(n!\right)}^{L}$$

*neurons and*

$$\left(L+1\right)n^{2}{\left(n!\right)}^{L}$$

*connections between them.*


The constructed network memorizes products in all *L*-tuples of permutation matrices, recognizes the input *L*-tuple of permutations, and sends the product to the output.

**Example** **1.**
*Consider n=2,L=3 (Figure 5). Then*

$$\pi_{0}=\left(\begin{array}{cc} 1 & 2\\ 1 & 2 \end{array}\right),\pi_{1}=\left(\begin{array}{cc} 1 & 2\\ 2 & 1 \end{array}\right),P_{0}=\left(\begin{array}{cc} 1 & 0\\ 0 & 1 \end{array}\right),P_{1}=\left(\begin{array}{cc} 0 & 1\\ 1 & 0 \end{array}\right).$$


$$X_{1}=\left(\begin{array}{cc} x_{11}^{\left(1\right)} & x_{12}^{\left(1\right)}\\ x_{21}^{\left(1\right)} & x_{22}^{\left(1\right)} \end{array}\right),X_{2}=\left(\begin{array}{cc} x_{11}^{\left(2\right)} & x_{12}^{\left(2\right)}\\ x_{21}^{\left(2\right)} & x_{22}^{\left(2\right)} \end{array}\right),X_{3}=\left(\begin{array}{cc} x_{11}^{\left(3\right)} & x_{12}^{\left(3\right)}\\ x_{21}^{\left(3\right)} & x_{22}^{\left(3\right)} \end{array}\right).$$

*For example, we have $5=1\cdot {\left(2!\right)}^{2}+0\cdot \left(2!\right)+1$ for j=5; therefore,*

$$y_{5}=x_{1,\pi_{1}\left(1\right)}^{\left(1\right)}\cdot x_{2,\pi_{1}\left(2\right)}^{\left(1\right)}\cdot x_{1,\pi_{0}\left(1\right)}^{\left(2\right)}\cdot x_{2,\pi_{0}\left(2\right)}^{\left(2\right)}\cdot x_{1,\pi_{1}\left(1\right)}^{\left(3\right)}\cdot x_{2,\pi_{1}\left(2\right)}^{\left(3\right)}\cdot \prod_{\left(\alpha ,\beta \right)\neq (i,\pi_{a_{j,s}}\left(i\right))}\bar{x_{\alpha ,\beta }^{\left(\gamma \right)}}=$$


$$x_{1,2}^{\left(1\right)}\cdot x_{2,1}^{\left(1\right)}\cdot x_{1,1}^{\left(2\right)}\cdot x_{2,2}^{\left(2\right)}\cdot x_{1,2}^{\left(3\right)}\cdot x_{2,1}^{\left(3\right)}\cdot \bar{x_{1,1}^{\left(1\right)}}\cdot \bar{x_{2,2}^{\left(1\right)}}\cdot \bar{x_{1,2}^{\left(2\right)}}\cdot \bar{x_{2,1}^{\left(2\right)}}\cdot \bar{x_{1,1}^{\left(3\right)}}\cdot \bar{x_{2,2}^{\left(3\right)}}.$$

*We can write similar expressions for all other $y_{j}.$*

*In this case, we have $I_{0}=\left\{0,3,5,6\right\},$ $I_{1}=\left\{1,2,4,7\right\}$ and $M_{11}=\left\{0\right\},$ $M_{12}=\left\{1\right\},$ $M_{21}=\left\{1\right\},$ $M_{22}=\left\{0\right\},$ so*

$$z_{1,1}=z_{2,2}=y_{0}\vee y_{3}\vee y_{5}\vee y_{6},$$


$$z_{1,2}=z_{2,1}=y_{1}\vee y_{2}\vee y_{4}\vee y_{7}.$$



**Example** **2.**
*Let n=3,L=2 (Figure 6). Then*

$$\pi_{0}=\left(\begin{array}{ccc} 1 & 2 & 3\\ 1 & 2 & 3 \end{array}\right),\pi_{1}=\left(\begin{array}{ccc} 1 & 2 & 3\\ 1 & 3 & 2 \end{array}\right),\pi_{2}=\left(\begin{array}{ccc} 1 & 2 & 3\\ 2 & 1 & 3 \end{array}\right),$$


$$\pi_{3}=\left(\begin{array}{ccc} 1 & 2 & 3\\ 2 & 3 & 1 \end{array}\right),\pi_{4}=\left(\begin{array}{ccc} 1 & 2 & 3\\ 3 & 1 & 2 \end{array}\right),\pi_{5}=\left(\begin{array}{ccc} 1 & 2 & 3\\ 3 & 2 & 1 \end{array}\right)$$


$$P_{0}=\left(\begin{array}{ccc} 1 & 0 & 0\\ 0 & 1 & 0\\ 0 & 0 & 1 \end{array}\right),P_{1}=\left(\begin{array}{ccc} 1 & 0 & 0\\ 0 & 0 & 1\\ 0 & 1 & 0 \end{array}\right),P_{2}=\left(\begin{array}{ccc} 0 & 1 & 0\\ 1 & 0 & 0\\ 0 & 0 & 1 \end{array}\right),$$


$$P_{3}=\left(\begin{array}{ccc} 0 & 1 & 0\\ 0 & 0 & 1\\ 1 & 0 & 0 \end{array}\right),P_{4}=\left(\begin{array}{ccc} 0 & 0 & 1\\ 1 & 0 & 0\\ 0 & 1 & 0 \end{array}\right),P_{5}=\left(\begin{array}{ccc} 0 & 0 & 1\\ 0 & 1 & 0\\ 1 & 0 & 0 \end{array}\right).$$

·


$P_{0}$



P1



P2



P3



P4



P5











P0




P0



P1



P2



P3



P4



P5



P1




P1



P0



P3



P2



P5



P4



P2




P2



P4



P0



P5



P1



P3



P3




P3



P5



P1



P4



P0



P2



P4




P4



P2



P5



P0



P3



P1



P5




P5



P3



P4



P1



P2



P0



$$X_{1}=\left(\begin{array}{ccc} x_{11}^{\left(1\right)} & x_{12}^{\left(1\right)} & x_{13}^{\left(1\right)}\\ x_{21}^{\left(1\right)} & x_{22}^{\left(1\right)} & x_{23}^{\left(1\right)}\\ x_{31}^{\left(1\right)} & x_{32}^{\left(1\right)} & x_{33}^{\left(1\right)} \end{array}\right),X_{2}=\left(\begin{array}{ccc} x_{11}^{\left(2\right)} & x_{12}^{\left(2\right)} & x_{13}^{\left(2\right)}\\ x_{21}^{\left(2\right)} & x_{22}^{\left(2\right)} & x_{23}^{\left(2\right)}\\ x_{31}^{\left(2\right)} & x_{32}^{\left(2\right)} & x_{33}^{\left(2\right)} \end{array}\right).$$


*For example, we have 15=2·(3!)+3 for j=15, so*

$$\begin{array}{c} y_{15}=x_{1,\pi_{2}\left(1\right)}^{\left(1\right)}\cdot x_{2,\pi_{2}\left(2\right)}^{\left(1\right)}\cdot x_{2,\pi_{2}\left(3\right)}^{\left(1\right)}\cdot x_{1,\pi_{3}\left(1\right)}^{\left(2\right)}\cdot x_{2,\pi_{3}\left(2\right)}^{\left(2\right)}\cdot x_{2,\pi_{3}\left(3\right)}^{\left(2\right)}\cdot \prod_{\left(\alpha ,\beta \right)\neq (i,\pi_{a_{j,s}}\left(i\right))}\bar{x_{\alpha ,\beta }^{\left(\gamma \right)}}=\\ x_{1,2}^{\left(1\right)}\cdot x_{2,1}^{\left(1\right)}\cdot x_{3,3}^{\left(1\right)}\cdot x_{1,2}^{\left(2\right)}\cdot x_{2,3}^{\left(2\right)}\cdot x_{3,1}^{\left(2\right)}\cdot \bar{x_{1,1}^{\left(1\right)}}\cdot \bar{x_{1,3}^{\left(1\right)}}\cdot \bar{x_{2,2}^{\left(1\right)}}\cdot \bar{x_{2,3}^{\left(1\right)}}\cdot \bar{x_{3,1}^{\left(1\right)}}\cdot \bar{x_{3,2}^{\left(1\right)}}\cdot \\ \bar{x_{1,1}^{\left(2\right)}}\cdot \bar{x_{1,3}^{\left(2\right)}}\cdot \bar{x_{2,1}^{\left(2\right)}}\cdot \bar{x_{2,2}^{\left(2\right)}}\cdot \bar{x_{3,2}^{\left(2\right)}}\cdot \bar{x_{3,3}^{\left(2\right)}}. \end{array}$$


*We can write similar expressions for the other $y_{j}.$*

*In this case, we have*

$$M_{11}=\left\{0,1\right\},M_{12}=\left\{2,3\right\},M_{13}=\left\{4,5\right\},$$


$$M_{21}=\left\{2,4\right\},M_{22}=\left\{0,5\right\},M_{23}=\left\{1,3\right\},$$


$$M_{31}=\left\{3,5\right\},M_{32}=\left\{1,4\right\},M_{33}=\left\{0,2\right\}$$

*and*

$$I_{0}=\left\{0,7,14,22,27,35\right\},I_{1}=\left\{1,6,16,20,29,33\right\},I_{2}=\left\{2,9,12,23,25,34\right\},$$


$$I_{3}=\left\{3,8,17,18,28,31\right\},I_{4}=\left\{4,11,13,21,24,32\right\},I_{5}=\left\{5,10,15,19,26,30\right\}.$$


*Thus*

$$z_{1,1}=\underset{s\in I_{0}\cup I_{1}}{⋁}y_{s}=y_{0}\vee y_{7}\vee y_{14}\vee y_{22}\vee y_{27}\vee y_{35}\vee y_{1}\vee y_{6}\vee y_{16}\vee y_{20}\vee y_{29}\vee y_{33},$$


$$z_{1,2}=\underset{s\in I_{2}\cup I_{3}}{⋁}y_{s}=y_{2}\vee y_{9}\vee y_{12}\vee y_{23}\vee y_{25}\vee y_{34}\vee y_{3}\vee y_{8}\vee y_{17}\vee y_{18}\vee y_{28}\vee y_{31},$$


$$z_{1,3}=\underset{s\in I_{4}\cup I_{5}}{⋁}y_{s}=y_{4}\vee y_{11}\vee y_{13}\vee y_{21}\vee y_{24}\vee y_{32}\vee y_{5}\vee y_{10}\vee y_{15}\vee y_{19}\vee y_{26}\vee y_{30},$$


$$z_{2,1}=\underset{s\in I_{2}\cup I_{4}}{⋁}y_{s}=y_{2}\vee y_{9}\vee y_{12}\vee y_{23}\vee y_{25}\vee y_{34}\vee y_{4}\vee y_{11}\vee y_{13}\vee y_{21}\vee y_{24}\vee y_{32},$$


$$z_{2,2}=\underset{s\in I_{0}\cup I_{5}}{⋁}y_{s}=y_{0}\vee y_{7}\vee y_{14}\vee y_{22}\vee y_{27}\vee y_{35}\vee y_{5}\vee y_{10}\vee y_{15}\vee y_{19}\vee y_{26}\vee y_{30},$$


$$z_{2,3}=\underset{s\in I_{1}\cup I_{3}}{⋁}y_{s}=y_{1}\vee y_{6}\vee y_{16}\vee y_{20}\vee y_{29}\vee y_{33}\vee y_{3}\vee y_{8}\vee y_{17}\vee y_{18}\vee y_{28}\vee y_{31},$$


$$z_{3,1}=\underset{s\in I_{3}\cup I_{5}}{⋁}y_{s}=y_{3}\vee y_{8}\vee y_{17}\vee y_{18}\vee y_{28}\vee y_{31}\vee y_{5}\vee y_{10}\vee y_{15}\vee y_{19}\vee y_{26}\vee y_{30},$$


$$z_{3,2}=\underset{s\in I_{1}\cup I_{4}}{⋁}y_{s}=y_{1}\vee y_{6}\vee y_{16}\vee y_{20}\vee y_{29}\vee y_{33}\vee y_{4}\vee y_{11}\vee y_{13}\vee y_{21}\vee y_{24}\vee y_{32},$$


$$z_{3,3}=\underset{s\in I_{0}\cup I_{2}}{⋁}y_{s}=y_{0}\vee y_{7}\vee y_{14}\vee y_{22}\vee y_{27}\vee y_{35}\vee y_{2}\vee y_{9}\vee y_{12}\vee y_{23}\vee y_{25}\vee y_{34}.$$



## 4. Deep Neural Network Solution

The calculation of the matrix $Z=X_{1}\cdot X_{2}\cdot \ldots \cdot X_{L}$ can be performed using a deep learning network, multiplying sequentially: $Y_{1}=X_{1},$ $Y_{k}=Y_{k-1}\cdot X_{k},$ (k=2,…,L). Then, $Z=Y_{L}.$ The network diagram is shown in the Figure 7.

Let $X_{k}=\left(x_{ij}^{\left(k\right)}\right),$$Y_{k}=\left(y_{ij}^{\left(k\right)}\right).$ Then,
$$y_{ij}^{\left(k\right)}=y_{i1}^{\left(k-1\right)}\cdot x_{1j}^{\left(k\right)}⊕y_{i2}^{\left(k-1\right)}\cdot x_{2j}^{\left(k\right)}⊕\ldots ⊕y_{in}^{\left(k-1\right)}\cdot x_{nj}^{\left(k\right)}.$$
To calculate the entries of matrices, we use the conjunction and addition modulo 2.

**Theorem** **3.**
*For travel maze problems F(n,L), there is a deep neural network that has a depth of L,*

$$\left(2L-1\right)n^{2}$$

*neurons and*

$$2\left(L-1\right)n^{3}$$

*connections between them.*


## 5. Neural Network for r-Bounded Problem

The travel maze problem is called *r*-bounded (0≤r≤n−1) if the inequality $|\pi_{j_{k}}\left(i\right)-i|\leq r$ holds for all i=1,…,n; k=1,…,L. This means that the corresponding permutation matrices are banded matrices with the bandwidth r+1, i.e., $x_{ij}=0$ for |i−j|≥r+1.

Let *A* be a banded matrix of the bandwidth r+1. The maximum number of nonzero entries in an arbitrary row of *A* is at most 2r+1. The number of nonzero entries in *A* is at most
$$N_{r}=n^{2}-\left(n-r\right)\left(n-r-1\right)=n\left(2r+1\right)-\left(r^{2}+r\right).$$
If *A* and *B* are banded matrices of the bandwidth r+1 and t+1, respectively, then the product AB is a banded matrix of bandwidth r+t+1.

**Theorem** **4.**
*For r-bounded travel maze problems F(n,L), there is a shallow neural network that has a depth of 3,*

$$L\cdot N_{r}+n^{2}+{\left(n!\right)}^{L}$$

*neurons and*

$$\left(L\cdot N_{r}+n^{2}\right)\cdot {\left(n!\right)}^{L}$$

*connections between them.*


**Theorem** **5.**
*For r-bounded travel maze problems F(n,L), there is a deep neural network that has a depth of L,*

$$L\cdot N_{r}+\sum_{i=2}^{L}N_{ir}\;\;\mathrm{neurons}\;\mathrm{if}\;\;Lr\leq n-1,$$


$$L\cdot N_{r}+\sum_{i=2}^{\left[\frac{n-1}{r}\right]}N_{ir}+n^{2}\cdot \left(L-\left[\frac{n-1}{r}\right]\right)\;\;\mathrm{neurons}\;\mathrm{if}\;\;Lr>n-1,$$


$$2\cdot \sum_{i=2}^{L}\left(ir+1\right)N_{ir}\;\;\mathrm{connections}\;\mathrm{between}\;\mathrm{neurons}\;\mathrm{if}\;\;Lr\leq n-1,$$


$$\mathrm{and}\;2\cdot \sum_{i=2}^{\left[\frac{n-1}{r}\right]}\left(ir+1\right)N_{ir}+n^{2}\cdot \left(2r+n+1\right)\cdot \left(L-\left[\frac{n-1}{r}\right]\right)\;\;\mathrm{connections}\;\mathrm{if}\;\;Lr>n-1.$$



## 6. Conclusions and Outlook

A shallow neural network combined from elementary Rosenblatt’s perceptrons can solve the travel maze problem in accordance with Rosenblatt’s first theorem.The complexity of the constructed solution of the travel maze problem for a deep network is much smaller than for the solution provided by the shallow network (the main terms are $2Ln^{2}$ versus ${\left(n!\right)}^{L}$ for the numbers of neurons and $2L^{3}$ versus $Ln^{2}{\left(n!\right)}^{L}$ for the numbers of connections).

The first result is important in the context of the widespread myth that elementary Rosenblatt’s perceptrons have limited abilities and that Minsky and Papert revealed these limitations. This mythology has penetrated even into the encyclopedic literature [8].

Original Rosenblatt’s perceptrons [1] (Figure 1) can solve any problem regarding the classification of binary images and, after minor modification, even wider. This simple fact was proven in Rosenblatt’s first theorem, and nobody criticised this theorem and proof. The universal representation property of shallow neural networks were studied in the 1990s from different points of view, including the approximation of real-valued functions [9] and the evaluation of upper bounds on rates of approximation [10]. Elegant analysis of shallow neural networks involved infinite-dimensional hidden layers [11], and upper bounds were derived on the speed of decrease of approximation error as the number of network units increases. Abilities and limitations of shallow networks were reviewed recently in detail [12].

Of course, a single *R*-element can solve only linearly separable problems, and, obviously, not all problems are linearly separable. Stating this trivial statement does not require any intellectual effort. Minsky and Papert [5] considered much more complex systems then a single linear threshold *R*-element. They studied the same elementary perceptrons that Rosenblatt did (Figure 1) with one restriction: *receptive fields of A-elements are bounded*. These limitations may assume a sufficiently small diameter of the receptive field (the most common condition) or a limited number of input connections of each *A*-neuron. Elementary perceptrons with such restrictions have limited abilities: if we have only local information, then we cannot solve such a global problem as checking the connectivity of a set or the travel maze problem with one glance. We should integrate the local knowledge into global criterion using a sequence of steps. This intuitively clear statement was accurately formalised and proved for the parity problem by Minsky and Papert [5].

Without restrictions, elementary perceptrons are omnipotent. In particular, they can solve the travel maze problem in the proposed form, but the complexity of solutions can be huge (Theorem 2). On the contrary, the deep network solution (Theorem 3) is much simpler and seems to be much more natural. It combines solutions from the one-step permutations locally, step by step, whereas the shallow network operates by all possible global paths. Restriction of the possible paths of travel by a bounded radius of a single step (Section 5) does not change the situation qualitatively. (The restricted problem is simpler than the original one. This should not be confused with the *A*-elements’ receptive field limitations proposed by Minsky and Papert [5] that complicate all problems.)

The second observation seems to be more important than the first one: the properly selected deep solutions can be much simpler than the shallow solutions. In contrast to the widely discussed huge deep structures and their surprising efficiency (see the detailed exposition of the mathematics of deep learning in [13]) the relatively small but deep neural networks are non-surprisingly effective for the solution of problems where local information should be integrated into global decisions, such as in the discussed version of the travel maze problem. These networks combine the benefits of the fine-grained parallel procession and the solutions of problems at a glance with the possibility of emulating logic of sequential data analysis when it is necessary. The important question in this context is: “How deep should be the depth?” [14]. The answer depends on the problem.

Comparing a *lower bound* on the complexity of shallow networks with an *upper bound* on the complexity of deep networks for travel maze problem will provide a much deeper result than our theorems. We have not yet been able to obtain the lower complexity bound of shallow networks for travel maze problems. In this paper, our goal was more modest: we compared the Rosenblatt’s first theorem “at work” with the obvious sequential deep algorithm of depth of order *L*. The solution of the simplified travel maze problem is equivalent to the computation of the product of *L* permutation matrices $X_{1},X_{2},\ldots ,X_{L}$. The algorithm of depth *L* is equivalent to the parenthesization $(\left(\ldots \left(\left(X_{1}\cdot X_{2}\right)\cdot X_{3}\right)\cdot \ldots \right)\cdot X_{L}$. Other parenthesizations can easily produce algorithms of lower depth of order $\log_{2}L$. For example, for eight matrices, we can group multiplications as follows: $\left(\left(X_{1}\cdot X_{2}\right)\cdot \left(X_{3}\cdot X_{4}\right)\right)\cdot \left(\left(X_{5}\cdot X_{6}\right)\cdot \left(X_{7}\cdot X_{8}\right)\right)$. It is obvious that the minimal depth achievable by parenthesization for products of *L* square matrices growths with *L* as $\log_{2}L$. (The theory of optimal parenthesization of rectangular matrix chain products is well-developed; see, for example, [15] and references there.) The *problem* is in the optimal computation of permutation matrix chain products with the depth independent of *L*. Is there any algorithm better than memorising the answers and the recognition of the input chains without any real computation of products? In other words, is there an algorithm that provides significantly more efficient solutions to the travel maze problem than the proof of Rosenblatt’s first theorem does?

Thus, the open question remains: are the complexity estimates sharp? How far are our solutions from the best ones? We do not expect that this problem has a simple solution because even for multiplication of n×n matrices, no final solution has yet been found, despite great efforts and significant progress (to the best of our knowledge, the latest improvement from $n^{2.37287}$ to $n^{2.37286}$ was achieved recently [16]).

Another open question might attract attention: analysing the original geometric travel maze problem (see Figure 2 instead of its more algebraic simplification presented in Figure 3). It includes many non-trivial tasks, for example, convenient discrete representation of the possible paths with bounded curvature, lengths and ends, and the constructive selection of ϵ-networks in the space of such paths for preparing the input weights of *A*-elements.

Analysis of more realistic cases with differential paths and a possibility of stepping back requires geometric and topological methods. Differential paths with bounded curvature *k* and lengths *l* form a compact set $Q_{kl}$ in the space of continuous paths equipped by the standard $C^{0}$ metrics ρ. A paths with its ε-vicinity form a strip that we see in Figure 2a. Each strip has its smooth mean path. We assume that the strips do not intersect beyond small neighborhoods of several isolated points of mean paths intersections. The ε-network in the set $Q_{kl}$ and metrics ρ is finite. For an empirically given path, we have to find the closest paths from this ε-network. The additional challenge is in the discretization of the input information. We should present the paths as the broken lines on a grid. The selection of the grid should allow us to resolve the intersections. For reliable recognition, an assumption of the minimal possible angle α of mean path intersections or self-intersections is needed. After that, one can use the logic of Rosenblatt’s first theorem and produce an elementary perceptron for solving the travel maze problem in this continuous setting. The perceptron construction and weights will depend on *k*, *l*, ε, and α. The detailed solution is beyond the scope of our paper.

The travel maze problem is discussed as a prototype problem for mobile robot navigation [17,18]. There exist universal “Elastic Principal Graph” (ElPiGraph) algorithms and open access software for the analysis of the raw image [7]. These algorithms were applied to the very new areas of data mining: they were used for revealing disease trajectories and dynamic phenotyping [19] and for the analysis of single-cell data for outlining the cell development trajectories [20] The travel maze problem in its differential settings (Figure 2a) was used as a benchmark for the ElPiGraph algorithm that untangles the complex paths of the maze [7].

The complexities of functions computable by deep and shallow networks used for the solution of the classification problem were compared for the same complexity of networks [21]. The complexities of the functions were measured using topological invariants of the superlevel sets. The results seem to support the idea that deep networks can address more difficult problems with the same number of resources.

The problem of effective parallelism pretends to be the central problem that is being solved by the whole of neuroinformatics [22]. It has long been known that the efficiency of parallel computations increases slower than the number of processors. There is a well known “Minsky hypothesis”: the efficiency of a parallel system increases (approximately) proportionally to the logarithm of the number of processors; at the least, it is a concave function. Shallow neural networks pretend to solve all problems in one step, but the cost for that may be an enormous number of resources. Deep networks make possible a trade-off between resources (number of elements) and the time needed to solve a problem since they can combine the efficient parallelism of neural networks with elements of sequential reasoning. Therefore, neural networks can be a useful tool for solving the problem of efficient fine-grained parallel computing if we can answer the question: how deep should the depths be for different classes of problems? The case study presented in our note gives an example of a significant increase in efficiency for a reasonable choice of depth.

## Figures and Tables

**Figure 1 entropy-24-01635-f001:**
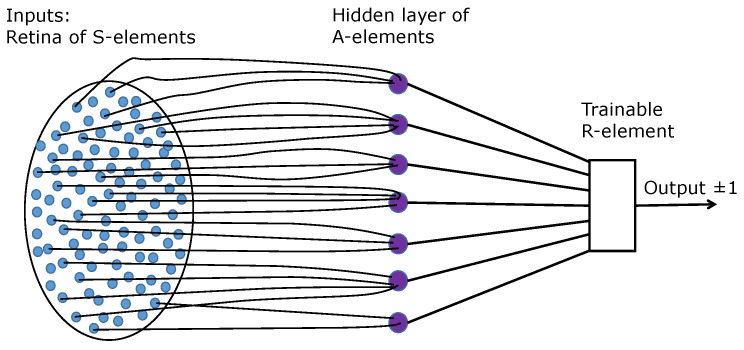
Rosenblatt’s elementary perceptron (re-drawn from the Rosenblatt book [1]).

**Figure 2 entropy-24-01635-f002:**
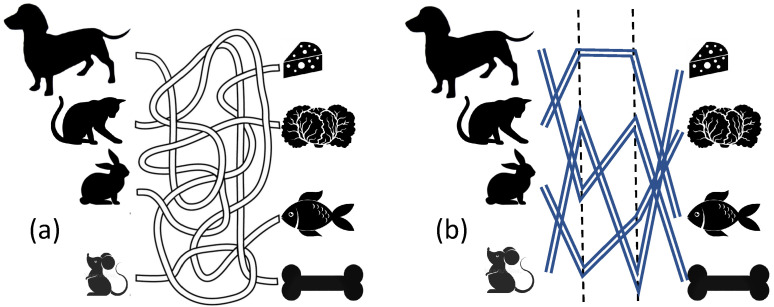
Have we chosen the right delicacies (**right**) for our guests (**left**)? (**a**) A prototype travel maze problem. (**b**) A simplified form of the problem with piece-wise linear paths for further formal description (Section 2). Complexity depends on the number of guests and the number of links in a path.

**Figure 3 entropy-24-01635-f003:**
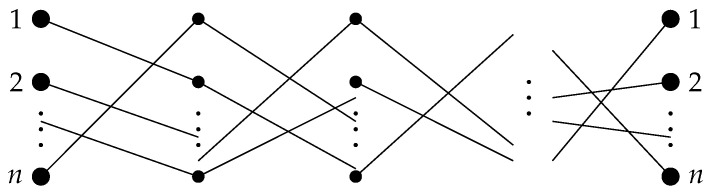
Game diagram with *L* stages (a formalized and simplified version of the travel maze problem).

**Figure 4 entropy-24-01635-f004:**
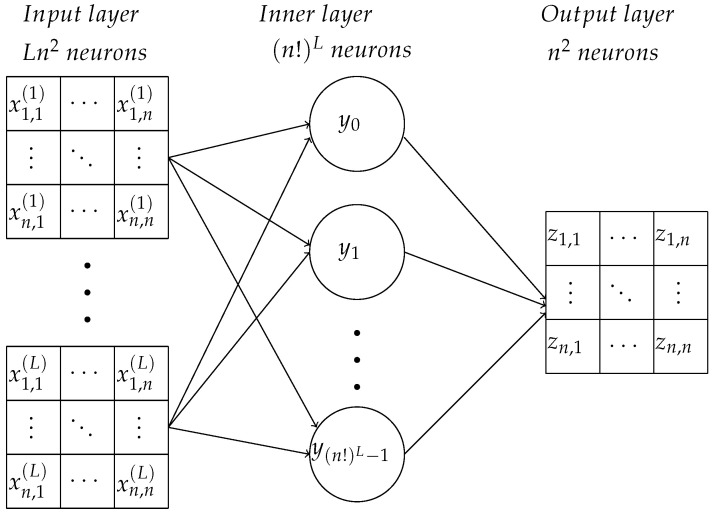
A shallow (fully connected) neural network for the travel maze problem (Figure 3). It differs from the classical elementary perceptron (Figure 1) by $n^{2}$ output neurons instead of one and can be considered as a union of $n^{2}$ elementary perceptrons with joint retina and hidden layer of *A*-elements.

**Figure 5 entropy-24-01635-f005:**
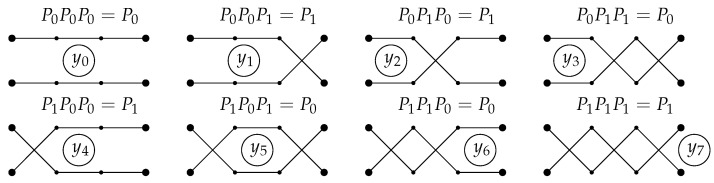
The case n=2,L=3.

**Figure 6 entropy-24-01635-f006:**
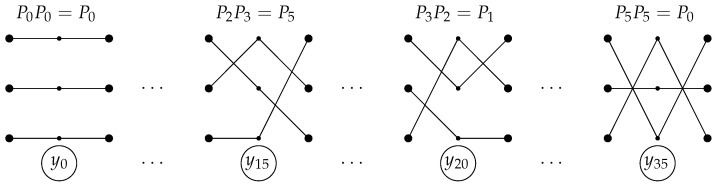
The case n=3,L=2.

**Figure 7 entropy-24-01635-f007:**
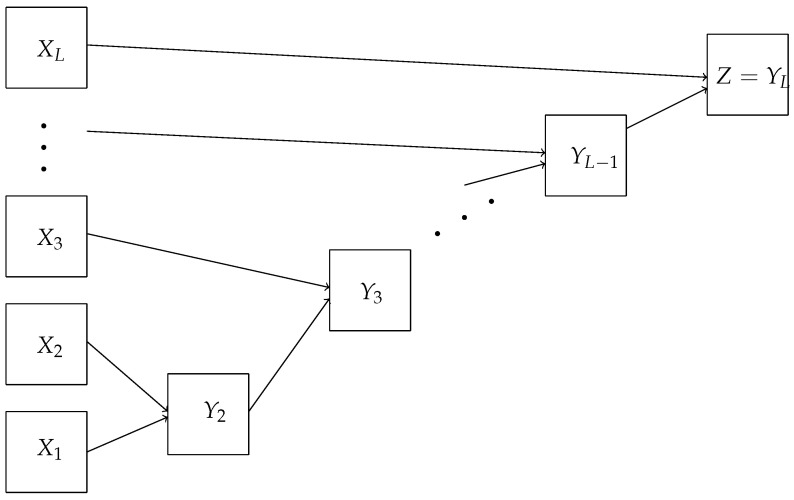
A deep neural network diagram for simplified travel maze problem.

## Data Availability

Not applicable.

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
