# Peer review of "Rosenblatt’s First Theorem and Frugality of Deep Learning"

_entropy, 2022, doi:10.3390/e24111635_

Round 1

Reviewer 1 Report

The paper contributes to important current problems of neurocomputing.
It
compares complexity requirements of deep and shallow networks for
computing the same class of functions. However, to be publishable,
rewriting improving clarity and mathematical rigor is necessary.
Formal definitions should be added (in particular the class of functions
modeling the travel maze problem should be formally defined as a family
of functions from {1,..,n} to {1,..,n} formed by composition of L
permutations parameterized by n and L. Theorems should be formulated
rigorously with quantifiers and formal notation of classes of functions
involved. The results are formulated so vaguely that it is not clear
whether results concerning shallow networks (section 3) represent a
lower bound or an upper bound on network complexity for the given class
of functions. Results comparing shallow and deep networks are only
interesting when a ``large’’ lower on complexity of shallow networks is
compared with a ``small’’ upper bound holding for deep networks.

Author Response

The authors are extremely grateful to the reviewers for their careful reading
of the manuscript and suggesting making changes, which greatly improved the
paper.

Reviewer 2 Report

The manuscript by Kirdin, Sidorov and Zolotykh "Rosenblatt’s First Theorem and Frugality of Deep Learning" presents a case study on comparing the elementary perceptron proposed by Rosenblatt in 1950s and a deep neural network, with respect to solving a particular task, the travel maze problem, characterized by the complex non-local connectivity of the input image. The complexity analysis of both solutions have been performed and compared.

The manuscript is very well written, its message is clear and all analyses made are rigorous. 

I was very excited to read the paper, since it clarifies the historical controversy connected with the Minsky and Papert's conclusions about "fundamentally limited Rosenblatt's perceptron" that caused, among other reasons, the "AI winter" in 1980s.

The manuscript clearly shows that unrestricted elementary perceptron is "omnipotent" while the Minsky's conclusions were about "restricted elementary perceptron" with unsurprising consequences.

As such the manuscript represents a great piece of pedagogy for the new generation of AI researchers. Besides this, the methodology of complexity estimation developed in the paper is a valuable novel contribution. Therefore, I am convinced that this manuscript must be published.

I have only few relatively minor demands to clarify the manuscript message:

1) The authors propose a simplified formalization (Figure 2b) of the real-life travel maze problem (Figure 2a).  What will change in the conclusions and in the actual complexity estimates in a more realistic case, continuous and with a possibility of step backs, for example?

2) Can the authors discuss in slightly more formal details the relation between the Minsky and Pappert results (theorem) and the Rosenblatt's results (theorem)? 

3) "This should not be confused with the possible network limitations, that complicate all problems." statement was not completely clear to me. What network limitations are meant here?

4) Are there examples in the literature, of real-life problems related to image analysis of similar nature or similar type of complexity as in the proposed travel maze problem, and what are the existing AI models for solving them in a realistic setting? In other words, what does it take to actually solve the travel maze problem presented in Figure 2 (or, similar), starting from the "raw image"?

Author Response

(The authors gave the same response as above.)

Round 2

Reviewer 1 Report

The manuscript is now more clear. It is interesting and I am glad to recommend it for publication.